# Projections of the economic burden of care for individuals with dementia in mainland China from 2010 to 2050

Yixiang Huang[1,2,☯,¤]*, Xiande Li[1☯], Zifeng Liu[3☯], Jinhai Huo[4], Jianwei Guo[1], Yingying Chen[5], Yanmei Chen[6], Ruoling Chen[7]

**1** School of Public Health, Sun Yat-sen University, Guangzhou, China, **2** Guangdong Health Economics Association, Guangzhou, China, **3** Department of Clinical Data Center, The 3[rd] Affiliated Hospital, Sun Yat-sen University, Guangzhou, China, **4** Department of Health Services Research, Management and Policy, The University of Florida, Gainesville, Florida, United States of America, **5** Guangdong Hospital of Traditional Chinese Medicine, Guangzhou, China, **6** The First Affiliated Hospital, Sun Yat-sen University, Guangzhou, China, **7** Centre for Health and Social Care Improvement (CHSCI), Faculty of Education, Health and Wellbeing (FEHW), University of Wolverhampton Millennium City Building, Wolverhampton, United Kingdom

☯ These authors contributed equally to this work.
¤ Current address: Department of Health Management, School of Public Health, Sun Yat-Sen University, Guangzhou, P.R. China
* huangyx@mail.sysu.edu.cn

**Data Availability Statement:** There are no restrictions on sharing the minimal dataset we used for analysis in our paper. The minimal dataset required for analysis are shown in Table 2 in the

## Abstract

### Background

China has stepped into an era of aging society, where the impending considerable economic burden attributed to high prevalence of dementia in the elderly appears to be one of the most important health and social issues to deal with for the country. However, population-based quantification and projections for the economic burden of dementia in China are lacking for further health action and policy making.

### Objective

To estimate and predict the costs of managing dementia in the elderly population aged 60 and above from 2010 to 2050 in China.

### Methods

Data were collected from a six-province study (n = 7072) and other multiple sources for calculation of the economic burden of dementia. With the convincing data from published studies, we quantified and projected the costs attributed to dementia in China from 2010 to 2050.

### Results

The national cost of dementia in 2010 was estimated to be US$22.8 billion by the opportunity cost method and US$26.4 billion by the proxy method. In 2050, the costs would increase to US$372.3 billion by the opportunity cost method and US$430.6 billion by the proxy method, consuming 0.53% and 0.61% of China's total GDP, respectively. A series of

paper, as other data we used for analysis were subtracted from published literatures (as shown in Table 1 in the paper).

**Funding:** This work was supported by the National Social Science Fund of China (grant no. 18BGL218),and Research and Demonstration of Integrated Medical Care Model for Dementia and Hypertension Treatment and Rehabilitation Based on Artificial Intelligence (grant no. 202002020047). The five-province cohort studies of dementia in China were funded by the BUPA Foundation (Grants Nos. 45NOV06, and TBF-M09-05) and Alzheimer's Research UK (Grant No. ART/PPG2007B/2).

**Competing interests:** The authors have declared that no competing interests exist.

sensitivity analyses showed that the changes in the proportions of informal caregiving led to the most robust changes in the total burden of care for dementia in China.

## Conclusion

Dementia represents an enormous burden on China's population health and economy. Due to the changes in policies and population structure, policymakers should give priority to dementia care.

## Introduction

Dementia is the primary cause of disability among the elderly, and caring for the elderly with dementia leads to immense burdens on their families, society, and country [1, 2]. In 2016, 47 million people worldwide lived with dementia. This number is predicted to be about 131 million by 2050, due to the aging of the population [3, 4]. The growing number of elders suffering with dementia has a great impact on the global economy. In 2010, the worldwide cost attributed to dementia increased from US$604 billion, equal to 1.01% of the global Gross Domestic Product (GDP) in 2010, to US$818 billion, equal to 1.09% of the global GDP in 2015 [3, 4].

Limited cost-of-illness (COI) studies on dementia have been conducted in low- and middle-income countries (LMICs), where over 50% of patients with dementia reside. As the largest LMIC in the world, China is expected to have 16 million people with dementia by 2030, equal to about 20% of all dementia patients worldwide [3, 4]. This rapid rise in the number of dementia sufferers will potentially lead to shortages of professional care, and many of these patients will receive long-term non-professional care provided by their family members instead [5]. Therefore, maintaining the economic sustainability of caring for a large number of elderly patients with dementia has become a significant challenge.

Previous studies on the economic burden of dementia in China had several limitations [6–8]. First, they did not use a nationally representative study sample but rather sampled patients from hospitals, especially tertiary hospitals. Second, these studies focused mainly only on Alzheimer's-type dementia. Third, these studies neglected the costs associated with people suffering from undetected dementia [6, 9]. Therefore, the projected costs of dementia were likely to be biased when the only data used were collected from people with a definitive diagnosis from tertiary hospitals [5, 7].

Our study was designed to estimate and predict the costs of managing dementia in the elderly population aged 60 and above from 2010 to 2050 in China, using more accurate prevalence data and detailed patient costs, adjusted for characteristics of both patients and caregivers.

## Methods

### Data sources

We collected diagnostic information for dementia patients from a six-province study (n = 7072; three independent studies: the four-province study, the Hubei study, and the Anhui study), whose samples covered the central (Hubei), west (Shanxi), east (Shanghai, Anhui), north (Heilongjiang), and south (Guangdong) of China. The four-province study was conducted in 2008–2009 to study mental health in older adults aged ≥ 60 in China. A cluster-randomized sampling method was used to choose residential communities from Shanxi, Shanghai, Heilongjiang, and Guangdong provinces. In total, 4314 participants, with a response

rate of 93.8%, were included and completed the interview where data related to dementia were collected. The Hubei study, conducted in 2010–2011 as the extension of the four-province study, included 1001 participants with a response rate of 91.8% by means of the same protocol. The Anhui study was a follow-up study starting from 2001, and in its third-wave survey carried out during 2007–2009, a protocol similar to that used in the four-province study was adopted for data collection. Finally, 1757 participants with a response rate of 82.4% were included in the analysis.

Information on patients' costs was also collected from literature reviews. We searched PubMed, China National Knowledge Infrastructure (CNKI), and Wanfang databases using keywords such as "dementia", "China", "economic burden", "caregiving", "formal care", and "informal care" for publications ranging from 2005 to 2015. Eight relevant studies with costs for specific data classifications were included [9–16]. Wage data were derived from the *China Statistical Yearbook* in 2010, public government statistics, and regional policy documents [17–19]. We then predicted the numbers of patients diagnosed with dementia from 2020 to 2050, using the Chinese Population Projection from the United Nations Population Information Network [4, 7]. A list of data we extracted for use from these sources is shown in detail in Table 1.

## Methods of measurement

We adopted a prevalence-based method, a more appropriate method than the incidence-based method, for estimating the burden of dementia. The six-province study survey, conducted by interviews, established a time-point for the prevalence of dementia, that is, the annual prevalence in terms of calculating person-years at risk. Several patients with prevalent dementia identified at any given time during the survey of the population (e.g., in January) would die later in the survey (e.g., in December), and in the meantime, several incident cases of dementia would be added for care-costing. Therefore, our data analysis study accounted for a mortality period in the study. Meanwhile, the 10/66 algorithm, a valid diagnostic method for use in

**Table 1. Sources of data projection.**

| Data for projection | Sources |
|---|---|
| Age-specific prevalence of dementia in China | 6-province study [7]. |
| Chinese population projection | World Population Prospects 2015 [20]. |
| Direct costs* | 6-province study [7]. |
| | Li M, et al [10]. |
| | Mould-Quevedo JF, et al [11]. |
| | Wang G, et al [9]. |
| | An C X, et al [12]. |
| Indirect costs* | Li M, et al [10]. |
| | Wang G, et al [9]. |
| | An CX, et al [12]. |
| | Wang H, et al [13]. |
| | Kuo LM, et al [14]. |
| | Wang J, et al [15]. |
| | Xiao LD, et al [16]. |
| Wage data | The *China Statistical Yearbook* in 2010 [17]. |

*For direct and indirect costs, we calculated the average cost after giving a weight to each study according to its sample sizes.

community-based studies in China, was used to identify the dementia patients [21]. With these data, we were able to conduct a prevalence-based COI analysis about dementia in China.

## Cost estimation

**Direct medical costs.** This included fees for medical resources such as inpatient and outpatient services with related medications and other therapies. For a more detailed estimation of direct medical costs, we collected data about medical services from 4 comparable studies with a total sample size of 2363 patients [9–12]. We calculated the final per capita cost with adjustments, by giving each study a weight based on its sample size and multiplying by the scale for patients receiving the medical services mentioned above.

**Direct non-medical costs.** Direct non-medical costs were divided into two parts: the costs of formal caregiving, and the costs of transportation and special equipment. A formal caregiver is defined as one who was hired as a professional in caregiving by hospitals, nursing homes, and other professional organizations. Similar to direct medical costs, only those patients diagnosed by specialists would have to pay for special equipment and transportation to and from the hospital for treatment for dementia. Thus, we estimated the average costs accordingly based on other published data [10, 12].

**Indirect costs.** Indirect costs were attributed to the loss of productivity by both patients and their informal caregivers [20, 22]. An informal caregiver was defined as a relative or friend without any employment relationship to the patient. We estimated informal caregiving by multiplying the annual average time of informal caregiving per patient, the estimated wages of informal caregivers, and the number of patients receiving informal caregiving as mentioned above. We obtained the data on caregiving time from existing research [9, 10, 12–26] and adopted two methods to estimate the wages of informal caregivers. The "proxy" method quantified the cost of an informal caregiver as the average fee for hiring a formal aide from the marketplace [2]. The "opportunity cost" method valued the hourly wage of on-the-job informal caregivers based on the national mean salary and wages of the remaining unemployed or retired caregivers, based on the national minimum hourly earnings scale [18]. We defined the minimum salary as 50% of the regional mean salary according to a government document released in 2004 [19].

Based on methods described in previous literature, we also estimated the burden of the entire population using only the economic burden of the diagnosed patient, regardless of patient type, for further comparisons with the estimates that accounted for patient type. Monetary costs were all converted into 2017 Chinese Yuan, at an annual discount rate of 3.5% [6]. In 2017, 6.97 Yuan were valued at 1 US dollar. For the prediction of the national economic burden in 2020–2050, we assumed that age-specific prevalence of dementia was equal for all ages, and that the resources required per patient were equal [2, 11].

## Sensitivity analysis

To test how the changes in different factors affected the economic burden of care for people with dementia in China, we conducted a sensitivity analysis and proceeded with the estimation by modifying the following parameters: (i) We assumed the percentage of undetected dementia to be 73.1% instead of 92.1%, which would directly affect the proportion of patients receiving direct medical services, including transportation and special equipment [5]; (ii) we assumed that the fees attributed to medical services would present an annual growth trend of 5% [6]; (iii) we assumed the proportions of recipients of formal and informal caregiving to be 4.9% and 86.0% rather than 7.6% and 30.2% [6]; (iv) when using the opportunity cost method, we valued the hourly wages of informal caregivers based on the national mean salary, without

any adjustments; (v) we estimated the cost of informal caregiving based on an average time of 6.3 hours per day instead of 15.4 [6]; (vi) we projected the total cost of care for dementia in the coming years with a discounted rate of 3% and 5% instead of 3.5%; and (vii) we adjusted the age-specific prevalence based on a systematic review [23].

## Results

Demographic characteristics of the dementia patients from the six-province study are shown in Table 2. In total, 377 dementia patients were identified, with a prevalence of 5.33%. In the six-province study, only 26 (6.9%) of the 377 patients with dementia were diagnosed and had

**Table 2. Demograhic characteristics of the dementia patients from the six-province study.**

| Variable | Dementia diagnosis | |
|---|---|---|
| | Undiagnosed (n = 351) | Diagnosed (n = 26) |
| Age(years) | | |
| 60~74 | 114 (32.5) | 8 (30.8) |
| 75~84 | 163 (46.4) | 14 (53.8) |
| 85~ | 74 (21.1) | 4 (15.4) |
| Gender | | |
| Male | 126 (35.9) | 8 (30.8) |
| Female | 225 (64.1) | 18 (69.2) |
| Urban-Rural | | |
| Urban | 138 (39.3) | 18 (69.2)** |
| Rural | 213 (60.7) | 8 (30.8) |
| Education level | | |
| High school and above | 14 (4.0) | 4 (15.4)* |
| Middle or primary school | 108 (30.8) | 6 (23.1) |
| Illiterate | 229 (65.2) | 16 (61.5) |
| Occupation (current/before retirement) | | |
| Manual (e.g. peasant) | 306 (85.8) | 19 (73.1) |
| Non-manual | 45 (14.2) | 7 (26.9) |
| Annual personal income (RMB, Yuan) | | |
| ≥20,000 | 51 (14.5) | 5 (19.2) |
| <20,000 | 300 (85.5) | 21 (80.8) |
| Household member average income (RMB, Yuan) | | |
| ≥20,000~ | 95 (27.1) | 9 (34.6) |
| <20,000 | 256 (72.9) | 17 (65.4) |
| Number of children | | |
| 0~3 | 106 (30.2) | 11 (42.3) |
| 4 or more | 245 (69.8) | 15 (57.7) |
| Family history of mental illness | | |
| No | 348 (99.1) | 23 (88.5)** |
| Yes | 3 (0.9) | 3 (11.5) |
| Overactive or underactive thyroid | | |
| No | 336 (95.7) | 21 (80.8)** |
| Yes | 15 (4.3) | 5 (19.2) |

*$P<0.05$
**$P<0.01$ for the univariate chi-square test.

received appropriate medical services. The remaining 351 (93.1%) of the 377 patients had never received any official dementia-related medical care. Thus, we defined their direct medical costs as 0. Only 25 (6.6%) of the 377 families had hired formal caregivers for patients, and we quantified that burden according to published data from the literature [9–12]. Only 100 (26.5%) of the 377 patients had ever received any type of informal caregiving.

## Estimated prevalence and numbers of patients with dementia

The population aged over 60 years numbered 175.9 million in 2010, and of those, the population with dementia numbered 7.5 million. It is expected that these numbers will grow to 491.9 million and 29.5 million in 2050. More detailed projections of the population and prevalence of dementia from 2010 to 2050 are shown in S1 Table.

## Estimated costs per person associated with dementia in 2010

Estimated average annual costs for the care of persons with dementia are shown in Table 3. In the opportunity cost method of valuing the informal caregivers' time, dementia was associated with an annual cost of US$3,044.00 per patient in 2010. The three major cost drivers are informal caregiving for patients (approximately US$2,534.40), accounting for 83.3% of the total, followed by direct medical costs (approximately US$261.10) and direct non-medical costs (approximately US$248.10). In the proxy method, the cost of informal caregiving was approximately US$3,002.10, and the total cost per patient increased to US$3,511.70 in 2010 (Table 3). Regardless of the types of dementia patients, we found that the direct medical cost per person with dementia was US$3,330.4 and the direct non-medical cost was US$3263.0, about 13 times the predicted value when the type of patient was considered. The indirect cost was US$8388.9 by the opportunity cost method or US$9,937.0 by the proxy method, about three times the predicted value after the type of patient was considered. Thus, the total economic burden was US$14,982.2 by the opportunity cost method or US$16530.3 by the proxy method, about five times the predicted value after the type of patient was considered (Table 3). Our study showed a cost range for dementia that covers that in the United States in 2010 [2] and similar results of total burden estimates from a previous study [3].

**Table 3. Annual cost of dementia per person in 2010 in China.**

|  | Annual cost per patient based on whether patients have been diagnosed (US$) | Annual cost per patient without consideration as to whether patients have been diagnosed (US$) |
|---|---|---|
| Direct medical cost | 261.6 | 3,330.4 |
| Direct non-medical cost | 248.1 | 3,263.0 |
| Formal caregiver | 205.2 | 2,716.8 |
| Transportation | 32.1 | 408.7 |
| Special equipment | 10.8 | 137.5 |
| Indirect cost |  |  |
| Informal caregiver (opportunity cost method) | 2,534.4 | 8,388.9 |
| Informal caregiver (proxy method) | 3,002.1 | 9,937.0 |
| Total (opportunity cost method) | 3,044.0 | 14,982.2 |
| Total (proxy method) | 3,511.7 | 16,530.3 |

NOTE: All the estimated costs were converted to United States dollar (US$) values in 2017, when one US$ was equivalent to 6.97 Chinese Yuan.

**Table 4. Annual cost of care for individuals with dementia in China from 2010 to 2050, differentiated by methods measuring indirect costs.**

| Cost of dementia, US$ billion | 2010 | 2010* | 2015 | 2015* | 2020 | 2020* | 2025 | 2025* | 2030 | 2030* | 2035 | 2035* | 2040 | 2040* | 2045 | 2045* | 2050 | 2050* |
|---|---|---|---|---|---|---|---|---|---|---|---|---|---|---|---|---|---|---|
| Direct medical costs | 2.0 | 25.0 | 2.9 | 36.4 | 4.1 | 51.8 | 5.8 | 73.8 | 8.6 | 109.8 | 12.7 | 162.0 | 17.3 | 220.5 | 23.6 | 300.0 | 32.1 | 408.3 |
| Direct non-medical costs | 1.8 | 24.5 | 2.7 | 35.6 | 3.9 | 50.8 | 5.4 | 72.3 | 8.3 | 107.6 | 12.1 | 158.7 | 16.4 | 216.0 | 22.4 | 294.0 | 30.5 | 400.0 |
| Formal caregivers | 1.5 | 20.4 | 2.2 | 29.7 | 3.2 | 42.3 | 4.5 | 60.2 | 6.8 | 89.6 | 10.0 | 132.1 | 13.6 | 179.9 | 18.5 | 244.7 | 25.2 | 333.1 |
| Transportation | 0.2 | 3.1 | 0.4 | 4.5 | 0.5 | 6.4 | 0.7 | 9.0 | 1.1 | 13.5 | 1.6 | 19.9 | 2.1 | 27.0 | 2.9 | 36.8 | 3.9 | 50.1 |
| Special equipment | 0.1 | 1.0 | 0.1 | 1.5 | 0.2 | 2.1 | 0.2 | 3.1 | 0.4 | 4.6 | 0.5 | 6.7 | 0.7 | 9.1 | 1.0 | 12.4 | 1.4 | 16.9 |
| Indirect costs | | | | | | | | | | | | | | | | | | |
| Informal caregivers (opportunity cost method) | 19.0 | 63.0 | 27.7 | 91.6 | 39.4 | 130.5 | 56.2 | 186.0 | 83.6 | 276.6 | 123.3 | 408.1 | 167.8 | 555.5 | 228.3 | 755.8 | 310.8 | 1,028.6 |
| Informal caregivers (proxy method) | 22.5 | 74.6 | 32.8 | 108.5 | 46.7 | 154.6 | 66.5 | 220.3 | 99.0 | 327.6 | 146.0 | 483.4 | 198.8 | 658.0 | 270.5 | 895.3 | 368.1 | 1,218.4 |
| Total (opportunity cost method) (percentage of national GDP, %) | 22.8 (0.49) | 112.5 (2.42) | 33.2 (0.41) | 163.6 (2.02) | 47.3 (0.37) | 233.1 (1.82) | 67.4 (0.36) | 332.1 (1.77) | 100.4 (0.39) | 494.0 (1.98) | 148.1 (0.43) | 728.8 (2.12) | 201.6 (0.45) | 992.0 (2.21) | 274.3 (0.48) | 1,349.8 (2.36) | 372.3 (0.53) | 1,837.0 (2.62) |
| Total (proxy method) (percentage of national GDP, %) | 26.4 (0.56) | 124.1 (2.63) | 38.3 (0.47) | 180.5 (2.22) | 54.6 (0.43) | 257.2 (2.03) | 77.8 (0.42) | 366.4 (1.98) | 115.8 (0.45) | 545.0 (2.12) | 170.8 (0.50) | 804.1 (2.35) | 232.5 (0.52) | 1,094.5 (2.45) | 316.4 (0.55) | 1,489.3 (2.59) | 430.6 (0.61) | 2,026.8 (2.87) |

NOTE: Data for national GDP refer to the projections of Goldman Sachs.

*Cost estimated without consideration as to whether patients have been diagnosed or not.

## Estimated costs in 2010 and projected costs from 2020 to 2050

The national costs attributed to dementia in 2010 totalled US$22.8 billion (by the opportunity-cost-based method) or US$26.4 billion (by the proxy-based method), equal to 0.49% or 0.56% of China's total GDP (Table 4). Assuming that the dedicated resources per patient remain constant over the year, the annual cost of care for those with dementia in China in 2020 will be double that in 2010, increasing to US$47.3 billion or US$54.6 billion. In 2050, the cost will be 16-fold that of 2010, from 0.53% (US$372.3 billion) to 0.61% (US$430.6 billion) of China's total GDP (Table 4, S2 Table, S1 and S2 Figs) [2, 6]. Regardless of the types of dementia patients, we found that the economic burden was US$112.5 billion (by the opportunity-cost-based method) or US$124.1 billion (by the proxy-based method) in 2010 and would reach from 2.62% (US$1.8 trillion) to 2.87% (US$2.0 trillion) of China's total GDP in 2050.

**Direct medical costs.** Estimated national direct medical costs attributed to dementia care were US$2.0 billion in 2010 and are projected to be US$32.1 billion in 2050. Regardless of the types of dementia patients, direct medical costs were US$25.0 billion in 2010 and are projected to be US$408.3 billion in 2050.

**Direct non-medical costs.** The direct non-medical costs totalled US$1.8 billion in 2010 and are projected to reach US$30.5 billion in 2050. Regardless of the types of dementia

patients, direct non-medical costs were US$24.5 billion in 2010 and are projected to be US$400.0 billion in 2050.

**Indirect costs.** In the opportunity cost method, the indirect costs were US$19.0 billion in 2010 and US$310.8 billion in 2050, about 83.3% of the total cost. If the proxy method was used, the indirect costs increased to US$22.5 billion in 2010 and US$368.1 billion in 2050, about 85.5% of total costs. Regardless of the types of dementia patients, the indirect costs were US$63.0 billion in 2010 and US$1.0 trillion in 2050 by the opportunity cost method and US$74.6 billion in 2010 and US$1.2 trillion in 2050 by the proxy method.

### Sensitivity analysis

Both the opportunity cost and proxy methods yielded similar results. The change in the proportion of those receiving informal caregiving, from 30.2% to 86.0%, led to the most robust impact, with a disease burden increased from 153.7% to 157.9%. In contrast, the change in the proportion of those receiving formal caregiving, from 7.6% to 4.9%, reduced the burden by no more than 2.4%. Our results also indicated that the variations in the percentages of undetected dementia, the hourly wages of informal caregivers, the discount rate of 5%, and age-specific prevalence would lead to an increased burden, and that those conservative changes in average time of informal caregiving and an annual discount rate of 3% would lead to a reduction (S1 and S2 Figs, S3 and S4 Tables).

## Discussion

Based on data from multiple sources, our estimates indicated that dementia care resulted in an annual burden ranging from US$3044.0 to US$3511.7 per case in China in 2010. The national cost consumed 0.49% to 0.56% of the national GDP in China in 2010 and represented from 3.77% to 4.37% of the global economic burden of care for those with dementia [3, 24]. Furthermore, the cost attributed to dementia would consume from 0.53% to 0.61% of the national GDP in 2050 [25]. Even though the caregiving system in China has not been completely developed, and many patients have not received proper care related to dementia, the burden we quantified is already enormous and worrisome. Predictably, dementia will be a sustained challenge to China in the future.

### Implications for policy development

All people with dementia should be diagnosed and receive appropriate care. However, low consultation rates and proportions of caregiving for patients with dementia have generated a gap between the actual burden and the ideal burden of costs for dementia care in China. The current patterns of caregiving and resources used for dementia care show some implications for policy development.

First, for patients with insufficient family support, medical insurance for long-term professional caregiving should be made more affordable, or free social assistance should be more accessible [11]. These conclusions were drawn based on the special period of the 'one-child' policy, which was established to control exploding population growth but has also led to an unhealthy situation where the only child has to cover most of the living costs of his/her parents. Since the health status of a parent with only one child would receive relatively less attention, dementia would likely go undiagnosed, and early treatment could not be implemented in time. Once a parent is found to be suffering from dementia, the child might not be able to afford the fees associated with medical services and professional caregiving, so the child would have to reduce working hours or even resign from his/her job to care for his/her parents informally. In 2014, more than 15 million family members and other unpaid caregivers

provided an estimated 17.9 billion hours of care to people with AD and other dementias, a contribution valued at more than US$217 billion [3]. In the six-province study, 25.0% of the informal caregivers had reduced their work-time or even resigned to look after their parents. This phenomenon will undoubtedly increase the indirect costs of care for those for dementia [21]. Thus, a larger range of medical insurance options and affordable professional caregiving for patients with dementia would be needed to reflect support from both government and society. Since the Chinese government puts forward five-year programs regularly, this could be one of the useful plans for the improvement of dementia care.

Second, publicity and education about dementia should be significantly strengthened. Due to the nationally low education level, many people consider dementia as a natural part of aging, which is often stigmatized. Thus, most dementia patients are either unaware of the availability of, or unwilling to turn to, professional assistance and are ultimately cared for by their friends or relatives. With improvements in the socio-economic environment and the implementation of the two-child policy, professional or formal caregiving would become more affordable and accessible. Thus, with more broad-based publicity and education, patients can receive timely formal assistance, and the availability of informal caregivers could reduce care time. These would decrease the indirect burden and make the ratio of the burden between formal and informal caregiving more balanced.

Third, standard training in healthcare for individuals with dementia through the national medical system should be considered. Even in some highly specialized tertiary hospitals in China, there are still no memory clinics, and most specialists have not been trained to care for individuals with dementia. In addition, many doctors are unwilling to make a diagnosis of dementia because they are unable to manage these individuals. Such facts will lead to longer periods of care and a lower quality of life of the patients' families [5]. According to the study by Brookmeyer and co-workers, over 40% of dementia patients worldwide require a higher level or longer period of care [26]. Therefore, the establishment of foundations for a system of long-term-care building and maintaining a sustainable and appropriately trained workforce is necessary to construct a long-term-care system such as called for by the WHO. Hence, there is a dire need for a standard healthcare system for dementia, including training in diagnostic procedures, medications, non-pharmacological interventions, and case management [5].

Finally, the gatekeeping system needs to be improved. The integrated care system in China is now under construction, so for most patients with dementia, the crucial gatekeeper for primary care is absent. Regarding the limited medical resources in China, a gatekeeper can keep patients from consuming unnecessary resources and, if there is a need, can also make it possible for patients with dementia to be served at specialist hospitals [5].

## For LMICs

As the largest LMIC in the world, China has benefited from an affluent labor force attributed to the booming population, at the expense of evolution into an accelerating aging society in a couple of decades. Likewise, for other typical LMICs that are developing following China's trajectory, their healthcare systems might be lagging and unable to support the mounting economic burden of dementia characteristic of an aging society [27]. Our projections and implications for policy development in China would therefore provide corresponding suitable references for other LMICs [3].

In addition to highlighting the need for better healthcare systems, the increased occurrence of dementia is making a significant demand on social care systems in LMICs and even developed countries [28, 29]. In particular, a lack of services (especially need-tailored ones), complexity of social care system regulations, as well as poor continuity of services have been cited as major

potential barriers to formal dementia care by patients, caregivers, and professionals [29]. Another significant factor contributing to the inefficient social care systems are the poor attitudes toward dementia–the disease, the affected patients, and their informal caregivers [29]. Given that these phenomena appear even in developed countries, causing a tremendous waste of social resources and a high economic burden, LMICs should realize the importance of establishing social care systems following top-level design standards and based on demand-driven principles.

## Strengths and limitations

Our study has the following strengths. First, it is the first population-based study quantifying and projecting the economic burden of dementia care in China. Second, compared with the *Diagnostic and Statistical Manual of Mental Disorders* (DSM) and the Mini-Mental State Examination (MMSE), we used diagnostic data based on the 10/66 algorithm, which is more reliable for population-based research in detecting undiagnosed dementia, especially for developing countries such as China [6, 30]. Third, when estimating the monetary cost of informal caregiving, we used an adjusted national average hourly wage instead of the mean wage or the minimum wage to make our estimation more rational and convincing [6, 9].

In the study by Jia et al., the costs of disease data for patients in hospitals were collected for the projection of economic burden of dementia care nationwide and reached the result that the average cost per person was US$19,144.36 [8]. This result was much higher than both the projections either considering or not considering diagnostic situations in our study, especially higher direct medical costs and direct non-medical costs [8]. Their study might have overlooked the fact that dementia patients diagnosed in hospitals were always of better socio-economic status or had more severe levels of dementia, adding selection bias to their studies. When these results were applied to national projections, those studies were all conducted based on an assumption that the per capita economic burden of care for patients with dementia, including undiagnosed patients, is the same as that of the hospital sample. This assumption is inconsistent with the current situation of dementia in China, and the projected costs might be overrated. In our study, only 7.9% of patients were diagnosed by specialists, while 37.8% of patients received caregiving. This result led to a lower expectation of dementia-attributed costs than reported in other studies, but also made our projections much closer to the truth. Although there would be other biases, such as recall bias, survivorship bias, and interview bias, they had less influence on the accuracy of information for chronic diseases including dementia. For the first time, our projection and sensitivity analysis indicated that informal care would be the source of the greatest burden, and that intervening informal care for dementia might be a key point for optimizing the future burden of dementia care.

Our analysis has several limitations. First, we had limited original data about the direct costs of care for persons with dementia. Second, when we forecast the economic burden, changes in both people's traditional thinking and national policies of the Chinese may have had an impact on our results. For example, the development of the two-child policy and the weakening effect of internal migration would greatly influence patterns of informal care, which might make our projections less robust. Third, our estimation neglected the intangible burden of care for dementia patients due to the lack of a standard approach for quantification. Moreover, stratified analyses for ages, types of dementia, severity levels, and other factors of dementia cost, especially those that might affect the types of care, were not conducted, due to a lack of original data and the limits of sample size. Also, limited references were suitable for use in the sensitivity analysis. Future research should enlarge the age ranges of participants, and the data about direct costs of the participants should also be collected and brought into the studies to enhance the reliability of the results.

In conclusion, care for individuals with dementia constitutes a substantial burden on China's economic system, and the situation is likely to worsen in the coming decades. Our study is of great significance not only for assisting people to realize the situation regarding care for people with dementia in China, but also for educating policymakers in other LMICs with booming aging populations about the importance of providing societal care for those affected with dementia.

## Supporting information

**S1 Table. Estimated numbers of people aged 60 years and above and people with dementia in China from 2010 to 2015.**
(DOCX)

**S2 Table. Annual cost of care for individuals with dementia in China from 2010 to 2050, differentiated by methods measuring indirect costs without discounting.**
(DOCX)

**S3 Table. Annual cost of care for individuals with dementia in China from 2010 to 2050, differentiated by methods measuring indirect costs without discounting.** NOTE: Data for national GDP refer to the projections of Goldman Sachs. *Cost estimated without consideration as to whether patients have been diagnosed or not.
(DOCX)

**S4 Table. Sensitivity analysis by the opportunity cost method.**
(DOCX)

**S5 Table. Sensitivity analysis by the proxy method.**
(DOCX)

**S1 Fig. The annual cost of care for individuals with dementia in China by the opportunity cost method.**
(DOCX)

**S2 Fig. The annual cost of care for individuals with dementia in China by the proxy method.**
(DOCX)

## Acknowledgments

Thanks are due to Xin Wang, Lijin Chen, and other members of Dr. Huang's team and to Weiju Zhou and other members of Dr.Chen's team for providing ideas for study design and the statistical analysis.

## Author Contributions

**Conceptualization:** Yixiang Huang, Xiande Li, Zifeng Liu, Ruoling Chen.

**Data curation:** Xiande Li, Zifeng Liu, Jinhai Huo.

**Formal analysis:** Yixiang Huang, Xiande Li, Zifeng Liu, Jinhai Huo, Jianwei Guo, Yingying Chen, Yanmei Chen, Ruoling Chen.

**Funding acquisition:** Yixiang Huang.

**Investigation:** Jinhai Huo, Ruoling Chen.

**Methodology:** Yixiang Huang, Xiande Li, Zifeng Liu, Jinhai Huo, Jianwei Guo.

**Project administration:** Ruoling Chen.

**Resources:** Ruoling Chen.

**Software:** Xiande Li, Zifeng Liu.

**Supervision:** Yixiang Huang, Ruoling Chen.

**Validation:** Yixiang Huang, Ruoling Chen.

**Visualization:** Yixiang Huang, Xiande Li, Zifeng Liu, Jianwei Guo.

**Writing – original draft:** Yixiang Huang, Xiande Li, Zifeng Liu, Yingying Chen, Yanmei Chen, Ruoling Chen.

**Writing – review & editing:** Yixiang Huang, Xiande Li, Zifeng Liu, Jinhai Huo, Jianwei Guo, Yingying Chen, Yanmei Chen, Ruoling Chen.

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
