## [Decision Letter · Decision Letter 0]

23 Dec 2021

PONE-D-21-25741Projections of the economic burden of care for individuals with dementia in mainland China from 2010 to 2050PLOS ONE

Dear Dr. Yixiang Huang,

Thank you for submitting your manuscript to PLOS ONE. After careful consideration, we feel that it has merit but does not fully meet PLOS ONE’s publication criteria as it currently stands. Therefore, we invite you to submit a revised version of the manuscript that addresses the points raised during the review process.

We look forward to receiving your revised manuscript.

Kind regards,

Ricky Chia Chee Jiun

Academic Editor

PLOS ONE

Journal Requirements:

Reviewers' comments:

Reviewer's Responses to Questions

5. Review Comments to the Author

Reviewer #1: Review report for the manuscript PONE-D-21-25741

Title: PLOS ONE Projections of the economic burden of care for individuals with dementia in mainland China from 2010 to 2050

In this study, the authors applied both opportunity cost and proxy methods to data from a six-province study and other multiple sources for calculation of the economic burden of dementia in the elderly Chinese population. As important results, the cost of dementia in 2010 was estimated to be below US$30 billion, while in 2050, the costs would increase to more than 10 times consuming 0.53% - 0.61% of China’s total Gross Domestic Product. Following sensitive analyses, the most robust changes in the total burden of care for dementia were attributed to the changes in the proportions of informal caregiving. The study therefore highlights dementia care as a priority to be addressed by policymakers. This is an important topic in health economics, well designed study and data analysis. The manuscript is well written, the text does not have flaws, the language is simple and easy to understand, appropriate illustrations are provided for findings and conclusion is based on the analyzed data. The methodology is detailed, data analysis is concise and relevant literature is documented. These kind of projections constitute an important implementation research mainly in the era of demographic expansion, as the stakeholders of LMICs concerned with increased number of elderly with dementia can exploit findings and plan for mobilization of necessary resources for addressing this burden. As minor observation, elaborate on the background of the abstract (L24-25). It does not state why it is necessary to quantify and project this economic burden. While projecting the population aged over 60 years by 2050 (L179-1982), could the impact of COVID-19 be factored, given than in some setups, it has significantly reduced this demographic group?

---

## [Author Response · Author response to Decision Letter 0]

4 Jan 2022

Response to Reviewer #1：

Thank you for your valuable and enlightening suggestions on our manuscript. We revised our manuscript as the following.

1. In the Abstract section (Line 24-25 in the original manuscript), we elaborated the necessity on our study objection as “China has stepped into an era of aging society, where the impending considerable economic burden attributed to high prevalence of dementia in the elderly appears to be one of the most important health and social issues to deal with for the country. However, population-based quantification and projections for the economic burden of dementia in China are lacking for further health action and policy making.” (Line 44-48 in the revised manuscript)

2. The impact of COVID-19 on the demographic group of dementia were an interesting and as well an absolutely valuable issue to discuss. The authors have discussed this issue at the start of the global pandemic of COVID-19. Generally, we considered COVID-19 little influence on the demographic group of dementia in China but uncertain influence in other LMICs for the following reasons:

(1) In mainland China, given that the COVID-19 outbreak were quite serious in 2020, and the COVID-19 were sporadic but also prevalent in 2021, there were 4,634 and 1,065 COVID-19 related deaths in 2020 and 2021, respectively [1,2]. Compared to the large population base of the 60-years and older in China (approximately 257.8 million in 2020 as estimated), the deaths due to COVID-19 were trivial in number to impact the demographic group of dementia. In this case, we expected COVID-19 has little impact on the reduction of the population size of dementia in China.

(2) Social isolation appeared to be one of the main social problems related to COVID-19 due to quarantine and travel restriction in China as well as the world. If the COVID-19 pandemic were expected to continue (such as the Omicron strain and further possible variation of the virus), we expected that COVID-19 would increase the incidence of dementia as study showed poor social engagement was associated with dementia risk [3]. In this case, we expected COVID-19 has an impact on increasing the population size of dementia in China, but the influence remains to be further studied.

(3) The third factor, would be the declining birth rate in mainland China. Research studying the relationship between COVID-19 and pregnancy or birth outcomes were emerging recently, but the impact of COVID-19 on birth remains to be verified further. Out study focus on the projection of the economic burden of dementia aged 60 and older from 2010 to 2050, which means the population we projected in the future were already born and aged 30 and older. In this case, we expected COVID-19 has no impact on the demographic group of dementia in China in our projection and estimation period.

(4) For LMICs outside China, we considered vaccination, screening policy, healthcare accessibility, and other anti-COVID-19 policy implementation intensity were factors influencing the impact of COVID-19 on the dementia population. However, as the COVID-19 pandemic were changing over days, the impact on dementia population is hard to estimate by now. Further study and statistics are needed to evaluate the impact in these LMICs.

Reference:

[1] https://voice.baidu.com/act/newpneumonia/newpneumonia/?from=osari_aladin_banner

[2] https://coronavirus.jhu.edu/map.html

[3] R Penninkilampi, AN Casey, MF Singh, et.al. The Association between Social Engagement, Loneliness, and Risk of Dementia: A Systematic Review and Meta-Analysis. J Alzheimers Dis. 2018;66(4):1619-1633. doi: 10.3233/JAD-180439.

[4] K Numbers, H Brodaty. The effects of the COVID-19 pandemic on people with dementia. Nat Rev Neurol. 2021 Feb;17(2):69-70. doi: 10.1038/s41582-020-00450-z.

---

## [Editor Report · Decision Letter 1]

12 Jan 2022

Projections of the economic burden of care for individuals with dementia in mainland China from 2010 to 2050

PONE-D-21-25741R1

Dear Dr. Yixiang Huang,

We’re pleased to inform you that your manuscript has been judged scientifically suitable for publication and will be formally accepted for publication once it meets all outstanding technical requirements.

Kind regards,

Ricky Chia Chee Jiun

Academic Editor

PLOS ONE
---

## [Editor Report · Acceptance letter]

25 Jan 2022

PONE-D-21-25741R1 

Projections of the economic burden of care for individuals with dementia in mainland China from 2010 to 2050 

Dear Dr. Huang:

I'm pleased to inform you that your manuscript has been deemed suitable for publication in PLOS ONE. Congratulations! Your manuscript is now with our production department. 

Kind regards, 

on behalf of

Dr. Ricky Chia Chee Jiun 

Academic Editor

PLOS ONE